# Pyrrolysine Aminoacyl-tRNA Synthetase as a Tool for Expanding the Genetic Code

**DOI:** 10.3390/ijms26020539

**Published:** 2025-01-10

**Authors:** Anastasia Dakhnevich, Alisa Kazakova, Danila Iliushin, Roman A. Ivanov

**Affiliations:** Biotechnology Department, Sirius University of Science and Technology, 354349 Sirius, Russia; a.dakhnevich@gmail.com (A.D.); severymonst@mail.ru (D.I.)

**Keywords:** pyrrolysine aminoacyl-tRNA synthetase, genetic code expansion, orthogonal pairs, tRNA engineering

## Abstract

In addition to the 20 canonical amino acids encoded in the genetic code, there are two non-canonical ones: selenocysteine and pyrrolysine. The discovery of pyrrolysine synthetases (PylRSs) was a key event in the field of genetic code expansion research. The importance of this discovery is mainly due to the fact that the translation systems involving PylRS, pyrrolysine tRNA (tRNA^Pyl^) and pyrrolysine are orthogonal to the endogenous translation systems of organisms that do not use this amino acid in protein synthesis. In addition, pyrrolysine synthetases belonging to different groups are also mutually orthogonal. This orthogonality is based on the structural features of PylRS and tRNA^Pyl^, which include identical elements, such as a condensed core, certain base pairs and the structural motifs of tRNA^Pyl^. This suggests that targeted structural changes in these molecules enable changes in their specificity for the amino acid and the codon. Such modifications were successfully used to obtain different aaRS/tRNA pairs that allow the incorporation of unnatural amino acids into peptides. This review presents the results of recent studies related to the correlation between the structure and activity of PylRS and tRNA^Pyl^ and the use of pyrrolysine synthetases to extend the genetic code.

## 1. Introduction

The first description of the mechanism of pyrrolysine incorporation into a protein was presented in 2004, when researchers led by Blight established how aminoacylation of pyrrolysine tRNA occurs with the involvement of PylRS [1]. In addition to the proteinogenic ones, there are more than 500 unnatural amino acids in nature that are not incorporated into proteins via ribosomal synthesis. A larger number of their variants can be chemically synthesized. The ability to incorporate ncAAs into a protein sequence in biological systems has significant advantages over classical synthesis, enzymatic conjugation and in vitro translation [2,3]. Each of the non-proteinogenic amino acids can potentially be used to improve the properties of naturally occurring protein molecules or to create new proteins [4]. The inclusion of ncAAs can result in the appearance of uncharacteristic chemical groups in recombinant proteins, which expands the possibilities of their use [2,5,6].

For example, the human PD-1 protein was modified by the incorporation of non-proteinogenic fluorosulfate L-tyrosine (FSY) at position 129 instead of alanine, which enhanced the antitumor effect by covalently binding PD-1 and PD-L1 with His69 in vivo [7]. Another example is the use of ncAAs in macrocyclic peptide drugs (MCPs). Zampaloni and colleagues reported on the possibility of optimizing an antibiotic by binding MCPs to it, solving the problem of *Acinetobacter baumannii* resistance to carbapenem antibiotics [8]. Another good example of ncAA application in protein therapeutics is tritrpticin [9]. Replacing Arg with Lys increased the selectivity of tritrpticin derivatives for cancer cells without causing a strong cytotoxic effect. Replacing Arg with unnatural Lys derivatives increases the selectivity of tritrpticin derivatives for cancer cells and also leads to rapid permeabilization of the cytoplasmic membrane. The introduction of ncAAs into the amino acid sequence is used to improve the properties of therapeutic protein drugs, such as biological activity, selectivity and stability as well as their half-life in the body [10,11,12].

Genetic code expansion (GCE) is a set of molecular biology techniques that enable the incorporation of ncAAs into protein sequences using ribosomal synthesis [13]. There are two main options for the implementation of GCE. The first strategy is the incorporation of ncAAs, which are structural analogs of natural amino acids, using modified endogenous aaRSs of the host cell [14,15]. This was achieved for both tyrosyl-tRNA synthetase (Eco TyrRS) and leucyl-tRNA synthetase (Eco LeuRS) of *E. coli* by modifying the anticodon of a cognate tRNA and performing successive rounds of selection for orthogonality and efficiency of ncAA incorporation [16,17]. This changes the specificity of a natural codon to enable the inclusion of ncAAs. This strategy is called sense codon reassignment. The second existing strategy is the site-specific incorporation of ncAAs in response to a stop or four-letter codon by exogenous aaRS and tRNA pairs [18,19].

There are several prerequisites for the site-directed insertion of ncAAs into proteins. A key element is the availability of a free codon, that is, the codon that does not normally encode any proteinogenic amino acid. Typically, site-directed GCE utilizes nonsense suppressor tRNAs that insert ncAAs in response to a stop codon [13]. The amber stop codon UAG is currently used as such a codon [20]. This codon is rare in most organisms and rarely involved in major proteins, which allows its use without affecting cell viability [21,22]. Another approach, the use of quadruplet codons, does not require recoding and offers a potential expansion from 64 to 256 codons [23,24].

In addition to a free codon, the presence of orthogonal pairs (aaRS)/tRNA is important [25] (Figure 1). Orthogonal aaRSs are those that do not interact with the proteinogenic amino acids and tRNA of the host cell, while they are complementary to the synthesis apparatus of the cell and selectively recognize one of the ncAAs as a substrate. The PylRS/tRNA^Pyl^ pair is often used in the context of site-specific GCE. The first distinguishing feature of the PylRS/tRNA^Pyl^ pair is that this system does not cross-react with endogenous aaRS or tRNA in either bacteria or eukaryotes, allowing its use in a variety of organisms [26]. The pyrrolysine synthetases are also characterized by orthogonality between the groups. This allows the use of synthetases of different classes and their tRNA to extend the genetic code in the production of proteins containing two or more ncAAs [27,28,29].

The second PylRS advantage is a large amino acid binding pocket in the active site and absence of an editing domain, which allows it to recognize a wide range of ncAAs [30,31]. The third feature is that the fact that tRNA^Pyl^ is a natural suppressor of the amber stop codon, and, unlike most aaRS enzymes, PylRS does not recognize the anticodon of tRNA^Pyl^ as an identity element [29,32], which allows the system to be used directly to recode the amber stop codon or to mutate the tRNA anticodon without compromising the efficiency of aminoacylation [33]. Mm-PylRS and Mb-PylRS are the most widely used enzymes of the PylSn–PylSc fusion group, which to date enable the site-specific incorporation of more than 100 ncAAs into various proteins [34].

The activity of synthetases and tRNA is determined by their structure. Amino acid substitutions in PylRSs can lead to a change in the affinity for pyrrolysine to another amino acid [26,35,36]. Such targeted substitutions are the subject of studies that advance synthetase engineering [37]. In addition to the structure of the synthetase, the structure of the associated tRNA also plays an important role. The specificity for ncAAs in the translation process is determined by the presence of identity elements, variable or, on the contrary, highly conserved structural elements of the molecule [27,38,39].

Modern approaches allow the incorporation of several non-proteinogenic amino acids into a protein during translation, which can significantly increase the functional capabilities of the new protein compared to proteins without or with only one ncAA. The potential applications of the inclusion of multiple ncAAs were highlighted in recent reviews [40,41]. The incorporation of multiple ncAAs can be achieved if the selected synthetases and tRNAs are orthogonal to the host aaRS/tRNA pairs as well as to other aaRS/tRNA pairs introduced to extend the genetic code [19,42,43].

The aim of this review is to summarize information related to the engineering of PylRS and tRNA^Pyl^ for the expansion of genetic code and the incorporation of non-proteinogenic amino acids into protein sequences. We decided to write this review after we had conducted our own research in the field of genetic code reprogramming. We are working on a project related to the site-directed mutagenesis of pyrrolysine aminoacyl-tRNA synthetases (PylRSs) using the phage-assisted continuous evolution (PACE) method. We have engineered several novel PylRS variants capable of integrating unnatural amino acids using the amber stop codon. We hope that this review will be useful for the community and provide valuable information for further research together with three recent reviews in this area [34,41,44].

In a recent review by Daniel L. Dunkelmann [41], authors discuss the design and optimization of PylRS/tRNA^Pyl^ pairs in terms of enzyme evolution to expand the genetic code in *E. coli* and other prokaryotic and eukaryotic systems. In contrast to our review, Dunkelmann et al. do not analyze the effects of specific mutations on the ability of PylRS to incorporate ncAAs, but rather provide a comprehensive summary of evolutionary methods for enzyme engineering. In addition, a large part of the review is devoted to applications of ncAAs that can currently be incorporated into proteins via biosynthesis. Dunkelmann et al. have convincingly shown that not only unnatural amino acids, but also small molecules that do not belong to amino acids, can be used for this purpose. It is important to emphasize that Dunkelmann et al. were the first to introduce new nomenclature for the groups of aminoacyl-tRNA synthetases and created triple- and even penta-orthogonal systems based on their mutual orthogonality.

The review paper by Nikolaj G Koch et al. [44], as the authors noted themselves, is primarily devoted to all the substrates that can be incorporated into proteins using PylRS and to applications of resulting proteins with a focus on therapeutic ones. Koch et al. summarize results of previous studies in the field of enzyme engineering methods and propose, in their opinion, the best system for the evolution of PylRS with improved in vivo efficiency for the recognition of new substrates. In contrast, our review focuses on specific mutations in different domains of PylRS that can improve the incorporation of ncAAs. Although the focus of our review is on PylRS, we also briefly address another major issue in orthogonal translation: the limited number of ncAAs that can be incorporated simultaneously and the need to increase the number of codons (apart from stop codons) used for the incorporation of ncAAs.

A review paper by Xuemei Gong et al. [34] provides a great overview of the key features of the PylRS/tRNA^Pyl^ pair that make this tool particularly suitable for genetic code expansion. In particular, their work focuses on important developments to improve the efficiency and (mutual) orthogonality of PylRS/tRNA^Pyl^ for the cotranslational and site-specific incorporation of multiple ncAAs. It should be noted that our review addresses a similar topic: the structural and functional properties of native and engineered PylRS/tRNA^Pyl^ pairs. However, Gong et al. address this topic from the perspective of new ncAA substrates and recent advances in the incorporation of different ncAAs using orthogonal pairs, as well as their optimization in terms of efficiency and substrate specificity.

Our team has a slightly different goal: to update information related to the effects of different mutations on the ability to incorporate unnatural amino acids into proteins using PylRS. In Section 2 we have summarized information on pyrrolysine synthetase and its interaction with tRNA^Pyl^. Section 3 is dedicated to PylRS groups such as “PylSn–PylSc”, “PylSn+PylSc” and “ΔPylSn” and a description of their structural features. In Section 4, we have carefully selected information on important mutations in the amino acid sequences of PylRS that improve the catalytic properties of the enzyme and increase its selectivity for certain groups of unnatural amino acids. One of the most important aspects of our review is the updating of the data on substitutions in the N-domain of the enzyme that not only improve its properties but can also be applied to the PylRS of other groups with an N-domain (Section 4.2). We then describe the tRNA^Pyl^ structure in Section 5. An important feature that distinguishes our work from other reviews is the chapter dedicated to tRNA engineering in the context of improving binding to ncAAs and mutant PylRSs (Section 6). Finally, Section 7 is devoted to the development of mutually orthogonal translation systems that allow the efficient insertion of multiple non-proteinogenic amino acids into the polypeptide sequence.

## 2. Pyrrolysine Aminoacyl-tRNA Synthetase

Pyrrolysine (Pyl) is the 22nd naturally occurring proteinogenic amino acid, which is only found in proteins of some anaerobic archaea and bacteria (along with selenocysteine). The special feature of these amino acids is that they are integrated into the peptide sequence by suppressing stop codons. For pyrrolysine, this is the UAG codon, the suppression of which depends on the ability of pyrrolysine aminoacyl-tRNA synthetase to aminoacylate the pyrrolysine tRNA [45]. At the same time, the integration of selenocysteine requires the presence of a specific SECIS element in the 3` untranslated region, which suppresses the UGA stop codon [46]. The archaea family *Methanosarcinaceae*, for example, produces Pyl and expresses Pyl-containing methyltransferases that ensure growth on methylamines [47]. In 1998, genes were discovered in the archaea *Methanosarcina barkeri* that enable the insertion of Pyl at the site encoded by the amber stop codon (TAG) [48]. As with the canonical amino acids, Pyl was found to be incorporated by the combined action of pyrrolysyl-tRNA synthetase (PylRS) and pyrrolysine tRNA (tRNA^Pyl^): PylRS specifically recognizes free Pyl and adds the amino acid to the 3′-hydroxyl of tRNA^Pyl^. Then, tRNA^Pyl^ incorporates Pyl into proteins in response to in-frame UAG codons during normal ribosomal protein synthesis [49]. Homologous methyltransferases, the Pyl biosynthesis and coding machinery were later discovered in other archaeal and bacterial species [47,50].

It should be noted that PylRS (*pylS*) and tRNA^Pyl^ (*PylT*) genes belong to the common Pyl gene cluster, which is adjacent to the methylamine methyltransferase gene cluster [47] (Figure 2). Three other genes also belong to the Pyl cluster: *pylB*, *pylC* and *pylD*, which are involved in the synthesis of Pyl from lysine (Lys). *PylB* encodes the S-adenosyl-L-methionine enzyme [51,52] and catalyzes the synthesis of (3R)-3-methyl-D-ornithine from Lys, whose carboxylate is then bound to the side chain amine of Lys by the product of the *pylC* gene, a member of the carbamoyl phosphate synthetase family, with the help of ATP. The resulting (3R)-3-methyl-D-ornithol-He-L-Lys is then oxidized by the NAD-dependent dehydrogenase gene product *pylD*, followed by simultaneous intramolecular condensation to form Pyl [53].

Crystal structure studies of PylRSs from different species have shown that the structure of the enzyme is similar to other class II aminoacyl-tRNA synthetases (aaRSs) and in most cases PylRS consists of two domains: a C-terminal domain (CTD) and an N-terminal domain (NTD). Like other class II aaRSs, PylRS exists as a dimer and has three loops in its active center. PylSc is the conserved catalytic domain, and its three-dimensional organization is similar to that of other class II synthetases: it, too, has a seven-stranded antiparallel β-sheet in the core surrounded by multiple α-helices that form a Rossmann fold for ATP binding [27]. A particular similarity is observed between PylRS and bacterial phenylalanyl-tRNA synthetase (PheRS), although the sequence identity is quite low (<30%) [30]. A phylogenetic analysis suggests that PylRS evolved from PheRS by duplication of the PheRS gene, and that the emergence of PylRS took place before the last universal common ancestor (LUCA) [33,44]. In addition, the hydrophobic pockets in which the cycloalkane group of Pyl and the phenyl group of phenylalanine (Phe) bind to the active site are organized in the same way in both enzymes. In both PylRS and PheRS, the inner surface of the binding pocket is formed by aromatic residues, and the pocket itself is closed by a loop that also contains an aromatic residue (Tyr384 in the case of PylRS and Phe260 in the case of PheRS). This loop also functions as a substrate-specific element in other class II synthetases for their related aminoacyl substrates [30]. However, a major difference between PylRS and PheRS lies in their amino acid composition; PylRS contains aromatic residues that form hydrogen bonds with the pyrrole ring of Pyl, whereas PheRS contains aromatic residues that cannot form hydrogen bonds due to the phenyl group of the PheRS substrate [28].

The most important amino acids of the PylRS binding pocket are Tyr384 and Asn346, which are responsible for specificity. Interaction with Tyr384 allows Pyl to adopt the correct orientation, a process mediated by hydrogen bonds between the γ-OH of tyrosine and the nitrogen of the pyrrole ring and the α-amino group [30]. After binding the correct substrate, the side chain of Tyr closes the hydrophobic pocket and thus completely encloses Pyl. Asn346 also forms hydrogen bonds with Pyl via the amide nitrogen of the side chain of N346 and the amide oxygen of the side chain of Pyl [54]. Although PylRS binds tightly to Pyl, this enzyme is able to aminoacylate tRNA^Pyl^ with Pyl analogs with different variations in the substrate side chain, especially when the variation is in the pyrroline region [55]. This is a unique situation, as other aminoacyl-tRNA synthetases are much more stringent in aminoacylation with derivatives of the cognate amino acid, and some even have an editing domain that hydrolyzes the complex of the wrong amino acid and tRNA. However, because PylRS recognizes the Pyl side chain primarily through relatively nonspecific hydrophobic interactions and lacks an editing domain, it is capable of aminoacylating an extremely wide range of amino acids [56].

As mentioned above, in addition to the C-terminal catalytic domain, PylRS enzymes from some organisms also possess a unique N-terminal tRNA-binding domain that appears to have no structural or sequential homology to any known RNA-binding protein, while having high structural similarity to the T- and variable loops of tRNA^Pyl^ [26].

The interaction between PylRS and tRNA^Pyl^ occurs in the same way as with other class II synthetases: PylRS approaches tRNA^Pyl^ from the major groove of the acceptor stalk, where the alpha-helix of tRNA-binding domain 1 and the C-terminal region together with the bulge domain of the neighboring subunit in the PylSc dimer form a binding site that accommodates the acceptor helix of tRNA^Pyl^ (Figure 3). This interaction directs the acceptor helix to the catalytic site where the enzyme, like other class II aaRSs, acylates the 3′-OH of the acceptor stalk [57]. A total of thirty-one residues of bacterial Dh-PylSc isolated from *Desulfitobacterium hafniense* are involved in hydrogen bonding or stacking interactions with seventeen bases in Dh-tRNA^Pyl^, with four of these residues belonging to the neighboring subunit of the PylSc dimer [58].

## 3. PylRS Enzyme Groups

The existing enzymes can be divided into the following three groups: “PylSn+PylSc”, fused “PylSn–PylSc” and “ΔPylSn” (Figure 4). The differences between the groups lie in the expression patterns of the two PylRS domains. The fused PylSc–PylSn group (first discovered in *Methanosarcina mazei* Mm-PylRS) has C and N domains that are cross-linked by a linker, whereas the PylSn group (first discovered in *Desulfitobacterium hafniense* Dh-PylRS) has CTD and NTD domains, which are two separate proteins that are combined into a single PylRS after translation. The last group, ΔPylSn (first discovered in *Candidatus Methanomethylophilus alvus* Ma-PylRS), lacks the tRNA-binding N-domain but is nevertheless highly active in *Escherichia coli* [28,47,59].

The linker part of the “PylSn–PylSc” fusion enzymes shows considerable variability between members of the *Methanosarcinaceae* family. In *Methanococcoides burtonii*, for example, the linker length is only 14 amino acids, whereas the linker in *Myceliophthora thermophila* consists of 72 amino acids [60]. The crystal structures of the isolated NTD PylRS from *M. mazei* solved so far show that this domain folds into a compact globule stabilized by a zinc ion and contacts the T- and variable loops of the tRNA^Pyl^, whereas the crystal structures of the CTD show that it interacts with the other side of the tRNA^Pyl^ [29] (Figure 3B). This close relationship between the NTD, the CTD and the variable loop of the tRNA^Pyl^ sterically explains the orthogonality of this group of PylRSs and their high specificity for the tRNA^Pyl^: the larger variable arm of canonical tRNAs would prevent their binding to the PylRS. Crystal structures reveal an elusive functional domain of pyrrolysyl-tRNA synthetase [27,29]. Furthermore, unlike aaRS, for most canonical amino acids there are no interactions between the tRNA^Pyl^ anticodon and the CTD or NTD of PylRS, facilitating the use of PylRS to encode non-natural amino acids (ncAAs) at different codons [58].

In the “PylSn+PylSc” group, the domains of two separate genes are expressed as two different proteins and then joined together to form PylRS (Figure 3B). The biological role of this separation is unclear, as is the role of the NTD in this group of PylRSs. It is known that Dh-PylRS is highly active in vivo, even when the RNA-binding domain is not expressed [61], suggesting that its expression is not required. It is possible that the biological function of the NTD is to recruit tRNA^Pyl^ to the catalytic domain, as suggested by the high binding affinity of the RNA-binding domain to tRNA^Pyl^ [26]. It has also been shown that the NTD not only recruits tRNA^Pyl^ but can also stabilize it in solution, facilitating its binding to the CTD and subsequent aminoacylation.

The third group “ΔPylSn” was recently discovered, and a characteristic of its representatives is that they completely lack the NTD and possess only an active CTD [62] (Figure 3B). ΔPylSn enzymes are often used when two different ncAAs need to be incorporated into proteins, as they are orthogonal to enzymes from the “PylSn–PylSc” fusion group [42]. ΔPylSns are present in recently described and largely uncultured methylotrophic methanogens of the seventh order called *Methanomassiliicoccales* [28,63,64,65]. Due to the lack of an RNA-binding domain and the atypical structure of the tRNA^Pyl^, it was initially unclear whether PylRS is active in these organisms. However, as in other known Pyl-incorporating organisms, the gene encoding MtmB carried an in-frame UAG codon, indicating the genetic encoding of Pyl [64]. The most promising enzyme of this group of PylRS was the enzyme Ma-PylRS, isolated from *Candidatus Methanomethylophilus alvus*, which was much more active in vivo than the widely used Mm-PylRS [59]. Subsequently, the activity of Ma-PylRS was confirmed by several studies [62,66,67], and the crystal structure of the enzyme was determined by two independent groups (PDB: 6EZD, 6JP2). Interestingly, despite the high activity of this enzyme in the absence of the RNA-binding domain, no remarkable features were found in the crystal structure: Ma-PylRS showed a high percentage of homology and structural similarity with Mm-PylRS and Dh-PylRS. Therefore, the high activity of the autonomous catalytic domain could be due to the peculiarities of the tRNA^Pyl^ of these organisms, which must independently adopt the correct orientation due to the absence of the NTD before the aminoacylation reaction.

Currently, PylRS genes are known to occur in 156 species of different anaerobic bacteria and archaea [33]. PylSc homologs are present in seventy-five species of archaea in eight phyla: *Euryarchaeota*, *Thermoplasmatota*, *Asgardarchaeota*, *Hydrothermarchaeota*, *Thaumarchaeota*, *Bathyarchaeota*, *Verstraetearchaeota* and *Korarchaeota*, with the majority (forty-seven) of these seventy-five identified genes belonging to the group of fused “PylSn–PylSc” enzymes. In addition, all organisms with fused “PylSn–PylSc” belonged to the *Methanosarcinaceae* family. At the same time, their relatives belonging to the same order *Methanosarcinales*, *Methermicoccus shengliensis* and *Methanomicrobia archaeon*, carry enzymes of the group “ΔPylSn” and “PylSn+PylSc”, respectively.

For 21 PylRSs found in archaea, it is not possible to determine the enzyme group precisely, as the genomes of these organisms were not completely elucidated. However, since PylSn could not be found in any of these genomes, it can be assumed that these PylRSs belong to “ΔPylSn”. This assumption is further supported by the fact that the PylSc genes present are highly similar to PylRSs from the “ΔPylSn” group. Most producers of these synthetases belong to the order of methanogenic archaea *Methanomassiliicoccales*, which inhabit the digestive system of animals [68]. In addition to *Methanomassiliicoccales*, enzymes of the ΔPylSn group are also found in the *Euryarchaeota* archaea as well as in *Bathyarchaeota* and *Thaumarchaeota*. Interestingly, enzymes of the “PylSn+PylSc” group are rare in archaea, as they were observed in only seven cases out of the seventy-five genomes analyzed [33].

A total of 81 PylRS genes were found in bacterial genomes. PylRSs were most frequently found in the phylum *Firmicutes*, especially in the order *Clostridiales* (41 genes). In addition to the *Firmicutes* phylum, pyrrolysine synthetases were also found in fourteen *Deltaproteobacteria*, two *Actinobacteria* and one *Spirochete*. In all cases, the bacterial PylRSs belonged to the group “PylSn+PylSc”.

Based on the taxonomic distribution of the Pyl-encoding organisms and recent phylogenetic analyses, it is assumed that the bacteria acquired the ability to incorporate Pyl by horizontal gene transfer from the *Methanomassiliicoccales*, which in turn inherited the ability to incorporate Pyl from the ancestral *Methanosarcinaea* [28,33,69,70].

## 4. Modification of the Structure of Native PylRS Variants to Extend the Range of Incorporated ncAAs

The first GCE studies were performed with native PylRS variants to incorporate a range of pyrrolysine derivatives into proteins [54]. Due to the proven low specificity of PylRS pyrrole ring recognition and the relatively large size of the hydrophobic pocket [29] to which this fragment belongs, the native enzyme could use pyrrolysine analogs in which a side chain was replaced by similarly sized structural components of a hydrophobic nature. Despite the fact that native PylRS has a broad spectrum of aminoacylated ncAAs, the efficiency and selectivity of the aminoacylation reaction is significantly lower for some amino acids than for others [30]. Therefore, the identification of a new or alternative evolution of a pair based on the existing one is an acute problem.

### 4.1. Changes to the C-Domain

By deciphering the crystal structures of PylRSs, amino acid residues in the active sites were identified, whose punctual changes made it possible to expand the repertoire of inserted ncAAs [27]. Thus, the following positions in the substrate binding pocket are considered important for the binding of the substrate of PylRS *M. mazei* as well as its homologs, Tyr306, Leu309, Asn346, Cys348 and Tyr384, forming a network of hydrogen bonds with Pyl [71,72,73,74].

It was shown that only two mutations, Asn346Ala and Cys348Leu (or Cys348Lys), are sufficient for Mm-PylRS to have high activity towards Phe [26,43,75,76]. The Asn346Ala mutation has been shown to significantly reduce the binding of PylRS to Pyl and remove the steric hindrance to phenylalanine binding [77]. When this mutation is introduced, a large empty space remains in the active site around the para-position of Phe, allowing the Asn346Ala PylRS mutant to be used to incorporate Phe derivatives with branched side chains. Substitution of Cys348 by an amino acid with a larger side chain, which takes the place of the Pyl pyrrolidine in the active site, leads to enhanced interactions with bound Phe. An important observation was that the substitution of Cys348 with a smaller amino acid residue enables the recognition of Phe derivatives with large para-substituents [35]. The double substitution of Cys348Ala and Asn346Ala (PylRS-AA) results in an enlarged active site pocket that provides greater structural flexibility for the para-substituted phenylalanine to bind to the active site [36]. Thus, the PylRS-AA mutant has a broad substrate spectrum that includes Phe derivatives with a variety of ortho-, meta-, and para-substituents as well as histidine derivatives, but is less effective at acylating Phe [34].

Mutations of Tyr384Phe or Tyr384Trp have also been shown to increase the activity of PylRS towards both Pyl and ncAAs [70,71,72]. It has been suggested that Tyr384 hydrogen bonding occludes the active site that serves to protect the unstable pyrrolisyladenylate when the cellular concentration of tRNA^Pyl^ is low [68] and forms specific hydrogen bonds with both the pyrrole nitrogen and the α-amino group of the substrate [30]. These data suggest that the hydrogen bond between the α-amino group of the substrate and Tyr384 is formed flexibly and is not essential for the catalytic activity of PylRS. Therefore, PylRS with a mutation at this position can recognize a larger number of substrates than the native enzyme. Another important position in PylRS to increase substrate specificity for ncAAs is Tyr306Ala. It was reported that the engineered PylRS mutant Tyr306Ala has an expanded binding pocket and that mutating the tyrosine residue to phenylalanine (Tyr384Phe) increases the efficiency of ncAA incorporation in vivo. Thus, such a double mutant can aminoacylate amino acids with branched side chains. Accordingly, proteins with branched cyclooctenes and octynes can be modified [78,79]. It has also been shown that this double mutant can incorporate furan-containing ncAAs that are significantly larger than Pyl [80].

It is also known that the substitution of a smaller alanine residue (Leu309Ala) for leucine at position 309 creates space for interaction between the hydrophobic and bulky residues at positions 348 and 306. In one study, the Cys348Phe and Leu309Ala mutations together resulted in an active site of AcKRS that was smaller and more hydrophobic than that of native PylRS, which explains why AcKRS was inactive toward the larger substrate Pyl [81].

All of these mutations improve the efficiency of incorporation of unnatural amino acids not only in the group of fused “PylSn–PylSc”, but also in other groups of the enzyme. For example, ISO4-G1 PylRS pairs have been developed [49]. Based on the sequence alignments, Ala121, Leu124, Tyr125, Met128, Asn165, Val16, Leu227 and Trp237 in the amino acid binding pocket of ISO4-G1 PylRS correspond to the previously described positions Ala302, Leu305, Tyr306, Leu309, Asn346, Cys348, Leu407 and Trp417 in Mm-PylRS, respectively. To accommodate bulky Lys derivatives such as ZLys, mAzZLys, N ε-(p-azidobenzyloxycarbonyl)-L-lysine (p AzZLys), N ε-(p-ethynylbenzyloxycarbonyl)-L-lysine (pEtZLys) and TCO*Lys, mutations at positions Tyr306 and Leu309 in Mm-PylRS were transferred to ISO4-G1 PylRS. As mentioned above, the Tyr125Ala mutation (Tyr306Ala in Mm-PylRS) enlarges the active site pocket in ISO4-G1 PylRS, which is then suitable for accommodating bulky ncAAs. In ISO4-G1 PylRS, the side chain of Met128 (Leu309 in Mm-PylRS) protrudes into the amino acid binding pocket, which reduces the size of the pocket compared to Mm-PylRS. Mutations of Leu and Ala at position 128 enlarge the interior of the active site pocket. Thus, the modifications in the active site of PylRS allowed the incorporation of ncAAs containing alkene, alkyne, azide, benzene, furan, and sulfur-containing groups into the proteins [54,76,82,83]. Interestingly, mutant PylRS was used to incorporate the ncAA N (epsilon)-acetyllysine, which is the most common type of posttranslational acylation of lysine in mammalian cells [84].

### 4.2. Changes to the N-Domain

Although most studies on the development of modified PylRS target the substrate binding pocket, mutations in other parts of the enzyme can also improve the incorporation of ncAAs. Since mutations in the N-terminal domain and the linker region between the domains do not affect AA substrate recognition, they can be transplanted into other previously identified PylRS variants to achieve similar improvements in activity. The T-loop and variable loop of tRNA^Pyl^ have hydrophilic interactions with PylRSn, and their truncation almost completely abolishes the binding specificity of PylRS to tRNA^Pyl^ [38]. Thus, a critical aspect in the group of fusion enzymes is the presence of a flexible loop between the CTD and NTD. The addition of a linker has been shown to provide additional flexibility to potentially regulate the interaction between the PylRS domains and alter the interface between PylRS and the tRNA^Pyl^, which in turn increases the efficiency of ncAA insertion [37]. As mentioned above, PylRSn is less soluble than PylRSc, so cleavage into two domains reduces the efficiency of the enzyme. However, it is known that Mm-PylRS is cleaved in bacterial cells after transcription in the linker part at positions Lys110 and Ala189. The mutation Pro188Gly keeps the NTD structure in the correct position, which disrupts protease recognition at Ala189 and prevents PylRS from being cleaved into two domains [85]. Therefore, in the fusion enzyme group, the linker modification enables the further regulation of interactions between domains and alters binding to the tRNA^Pyl^, which in turn increases the efficiency of aminoacylation [86].

The evolved PylRS carrying three mutations in the N-domain of PylRS Arg19His/His29Arg/Thr122Ser was also found to increase the efficiency of stop-codon repression upon the addition of BocK [87]. Another mutation that increases the catalytic properties of PylRS and affinity for tRNA^Pyl^ is the His62Tyr mutation. His62Tyr is predicted to disrupt two hydrogen bonds between PylRS and the T-loop fragments of tRNA^Pyl^, although it also makes new weaker contacts with the phosphate moiety Gly21 and base A20 [29].

In addition, the mutations Arg61Lys, His63Y and Ser193Arg are required to increase the efficiency of aminoacylation with the new amino acid. Arg61Lys and His63Tyr are located in PylRSn at the boundary between the NTD and the T-loop binding region of the tRNA. In particular, the guanidine side chain of Arg61 has a polar interaction with the phosphodiester backbone near 58A of tRNA^Pyl^ as well as with residue Arg52 within the NTD. In addition, the imidazole side chain of His62 forms hydrogen bonds with the side chain amino group of Lys85. The mutations Arg61Lys and His62Y can disrupt these interactions. The Ser193Arg mutation is located in the CTD and lies at the border of the tRNA D-loop binding region. Although Ser193 is relatively distant from the NTD, it does not interact with residues near the catalytic site. It has been proposed that this mutation locally expands the catalytic core of PylRS to allow ncAAs with shorter, bulkier side chains to bind to the tRNA^Pyl^ [37].

### 4.3. Comparisons of the Efficiency of Aminoacylation by Modified Synthetases

To compare the efficiency of aminoacylation and ncAA activation by native and modified PylRSs, enzyme kinetics studies were performed in a series of studies [81,88,89,90] (Appendix A). When aminoacylation was assessed, the resulting IFRS mutant, Asn346Ser/Cys348Ile, showed a significantly higher efficiency of UAG translation (~50% of WT sfGFP) than the native MmPlRS and MbPylRS, known variants of PylRS. Some meta-substituted phenylalanine analogs showed incorporation efficiencies of 30–68% [88]. In the same work, when testing the efficiency of ncAA activation, it was shown that for MmPlRS and MbPylRS Km is ~50 μM and the catalytic turnover during the formation of Pyl-AMP is 0.1–0.3 s^−1^. The AcKRS1 mutant is the most efficient catalyst with the substrate 3-trifluoromethyl-l-phenylalanine, its activity is 166-fold less efficient than the activity of PylRS WT with Pyl; however, Km for AckRS1 with AcK is 35 mM, and its kcat is only 10-fold lower than that of PylRS WT with Pyl. The best substrate for AcKRS3 is 3-I-Phe, but its activity is 288-fold lower compared to WT PylRS using Pyl as substrate. IFRS, which is five times more efficient than AckRS1 or AcKRS3 in the formation of 3-I-Phe adenylate, is 40 times less efficient in the aminoacylation of tRNA^Pyl^ compared to WT PylRS.

In another study, the effects of mutations in tRNA on aminoacylation efficiency were investigated. The results showed that the kcat and Km values for tRNA^Pyl^ and tRNA^Pyl^-opt do not differ and that there are no effects on catalytic efficiency and amino acid binding. Mutations in base pairs 2–71 and 3–70 increased the Km values for tRNA, indicating the weaker binding of tRNA to PylRS, whereas mutations in base pairs 6–67 and 7–66 did not significantly affect the binding of tRNA to PylRS [89].

In addition, the kinetic properties of PylRS when interacting with D-amino acids were investigated [90]. DFRS2 was found to exhibit pronounced aminoacylation activity with L-amino acids, while its Km values were significantly higher and kcat lower for D-amino acids. The relative efficiency of aminoacylation showed that the catalytic efficiency of DFRS2 with L-amino acids was 1.2–2.3 times higher and with D-amino acids 20–59 times lower than that of PylRS with BocK. When evaluating the efficiency of ncAA activation, the catalytic efficiency of DFRSc with respect to L-amino acids was found to be 15–32 times lower than that of PylRSc, and the low solubility of D-amino acids did not allow for the determination of kinetic parameters. When working with PACE, the kcat of the resulting tetramutant chPylRS (IPYE) improved 8.7-fold, the KM for the tRNA^Pyl^ substrate improved 5.7-fold and the catalytic efficiency of the evolved variant increased 45-fold compared to chPylRS [91].

Currently, orthogonal tRNA synthetases have moderate catalytic activity and are polyspecific for many non-canonical amino acids [15]. However, significant progress has already been reported for p-azido-PheRS by multiplexed automated genome engineering (MAGE) [92] and in PylRS by PACE or phage-assisted discontinuous evolution (PANCE) [91]. The generation of orthogonal pairs is also possible with other viruses, e.g., the adeno-associated virus, which provides the genes required for the directed evolution of tRNAs in mammalian cells. A group led by Jewel reported in 2023 and 2024 on the application of a virus-mediated directed evolution strategy [93]. This creates more functional and specific systems to increase the catalytic capacity and substrate specificity of PylRS.

## 5. Pyl tRNAs

For the successful translation of the genetic code, aaRSs must bind amino acids exclusively to their cognate tRNAs. The specificity of a tRNA for the corresponding aaRS is achieved by a set of individual features that are common to each type of tRNA. These features are referred to as identity elements [39]. Identity elements promote correct aminoacylation (determinants) or prevent incorrect aminoacylation (antideterminants). These elements include single nucleotides, single-stranded regions, base pairs or structural motifs. Such identity elements can occur in the L-shaped structure of tRNAs, but are most frequently found in the acceptor stem and in the anticodon loop. For example, base N73 in the acceptor stem and N35 and N36 in the anticodon loop are common identity elements of all tRNAs [39]. Thus, all aaRSs from *E. coli*, with the exception of glutamyl-tRNA synthetase and threonyl-tRNA synthetase, rely on the identity element N73 [94].

N73 is important for tRNA^Pyl^ in both archaea and bacteria. Thus, the discriminator base (G73) and the first base pair (G1:C72) were identified as critical elements for tRNA^Pyl^. These residues are highly conserved in organisms that incorporate Pyl into their proteins and they are also the ones that directly contact PylSc during aminoacylation [27]. Other identification elements (for Dh-tRNA^Pyl^) are the base pairs G10:C25 and A11:U24 and the base G9, all of which contact Dh-PylRS upon binding [38]. Interestingly, the identity elements in archaeal Mb-tRNA^Pyl^ are different: they are the base pair G51:C63, as well as U33 and A37, which are adjacent to the anticodon [32]. However, neither Mb-tRNA^Pyl^ nor Dh-tRNA^Pyl^ are sensitive to mutations in the anticodon and, if present, are able to bind Pyl.

Despite relatively low sequence similarity, Pyl tRNAs from bacteria and archaea *Methanosarcinaceae* expressed from species with PylRSs from the PylSn and PylSn–PylSc groups are structurally similar and have many unique features that distinguish them from other tRNAs (Figure 5). These features include the following: only one base between the acceptor stem and the D-stem; an anticodon stem containing six base pairs instead of five; a short variable loop of three bases; a D-loop of five bases instead of eight; and T54U and C56A mutations in the T54-Ψ55-C56 motif (where Ψ is pseudouridine) in the D- and T- loops, respectively [47]. For example, a closer examination of the nucleotide composition of Mb-tRNA^Pyl^ revealed that this tRNA contains only two modified nucleotides: 4-thiouridine at position 8 and 1-methyl-pseudouridine at position 50. It is possible that Mb-tRNA^Pyl^ contains pseudouridine, but the LC/ESI-MS experiments did not allow us to distinguish it from unmodified nucleotides by mass [95].

Bacterial tRNA^Pyl^ are more diverse in nucleotide composition than the highly conserved *Methanosarcineae* sequences; their sequence identity is only ~45% [96]. Examination of the structure of Dh-tRNA^Pyl^ revealed that this tRNA has a canonical L-form with a compacted core region [27]. The compact core of tRNA^Pyl^ is the result of non-standard tertiary base pairing, mainly due to the truncated D- and variable stems and the deletion of U8. Among canonical tRNAs, the U8:A14 linkage is highly conserved and plays an important role in the maintenance of the L-form [97]. Despite the denser structure, the acceptor and anticodon stems are located at positions similar to those of canonical tRNAs, allowing Dh-tRNA^Pyl^ to function normally during translation.

The enzymes of the ΔPylSn group are orthogonal to the other two PylRS groups and cannot recognize the Pyl-tRNAs specific to these groups [59], which is due to special identity elements in ΔPylSn Pyl-tRNAs that promote aminoacylation by only those PylRSs that contain only the CTD. As mentioned above, the three-nucleotide variable arm of the tRNA^Pyl^ is essential for aminoacylation by enzymes of the PylSn and PylSn–PylSc groups [59,98], while the longer variable arm carrying four or more nucleotides sterically collides with the NTD of PylRS and prevents the recognition of this enzyme by any other endogenous tRNA. However, PylRSs from the ΔPylSn group have no NTD and therefore this feature is not considered an identity element for these enzymes, and the addition of an extra nucleotide to the variable arm of the ΔPylSn-tRNA^Pyl^ creates an orthogonal translation system between the PylRS enzymes of the other two groups [59].

Furthermore, a search for tRNA identity elements interacting with Ma-PylRS from the ΔPylSn group showed that these tRNA elements together form a single structure [99]. This structure is formed by the absence of a base at position 8 and the presence of a wobble base pair G28:U42. However, as these two elements are not sufficient for the recognition of PylRS, they are complemented by G:C pairs (up to three) in both the acceptor and anticodon strains. The introduction of these identity elements into other Pyl-tRNAs that are not recognized by Ma-PylRS (from the PylSn–PylSc and PylSn groups) promoted the aminoacylation of Ma-PylRS and the subsequent protein translation [100].

Using MD modeling, it was shown that these elements cause rigidity in the molecule and promote the formation of a separate identity structure of the tRNA recognized by Ma-PylRS. Such a structure of Ma-tRNA^Pyl^ suggests the coevolution of PylRS enzymes of the ΔPylSn groups and their cognate tRNAs [33,100]. In contrast, in the PylSn–PylSc group, where the NTD is physically connected to the CTD via a linker, the cognate tRNA (Mm-tRNA^Pyl^) exhibited significantly lower stiffness with a looser structure resembling an obtuse angle rather than an L-shape, while in the PylSn group, where the two genes are separate, the cognate tRNA was stiffer and more L-shaped than in PylSn–PylSc, but not as stiff as the tRNA^Pyl^ from the ΔPylSn group, where there is no NTD.

The Pyl tRNAs within the ΔPylSn group are quite heterogeneous and contain additional structural changes. For example, in the ΔN tRNA^Pyl^, the acceptor and D-stems may be separated by one or two bases or have no base at all, the D-loop is shortened to four or three bases and the anticodon stem has additional unpaired residues that form a loop of five to seven bases [59].

## 6. tRNA^Pyl^ Engineering

Not only mutations in the catalytic and RNA-binding domains of PylRS, but also mutations in tRNAs can contribute to increasing the proportion of ncAAs incorporated into proteins. tRNA engineering can affect overall decoding efficiency through altered interactions with aminoacyl-tRNA synthetases and elongation factors, impaired ribosome accommodation or impaired mRNA decoding efficiency. In addition to modified tRNA^Pyl^ and other prokaryotic tRNAs, chimeric tRNAs based on bovine mtRNA^Ser^_UGA_ that are acylated by PylRS and act as orthogonal suppressors of amber can also be used [101].

Orthogonal tRNAs play an important role in the successful incorporation of ncAAs into GCE by directly influencing the efficiency of binding to aaRSs and the efficiency of aminoacylation, the thermodynamic and kinetic behavior of EF-Tu/GTP/ncAA-tRNA, and the correctness of amino acid incorporation into the protein. Since the PylRS/tRNA^Pyl^ pair is of archaeal origin, suboptimal recognition of tRNA^Pyl^ by the translational machinery in bacteria and eukaryotes leads to lower protein yield. However, this can be improved by using tRNA^Pyl^ variants engineered for better recognition by the host translation system. For example, Schultz et al. optimized the interaction between EF-Tu and tRNA by introducing mutations into the acceptor and T-strains of Mj-tRNA_Tyr_, such that the improvement in protein yield was between 175 and 2520%, depending on the ncAA used [102]. In another study addressing the EF-Tu binding problem for tRNA^Pyl^, the optimal mutations G7-C66, U49-A65 and G50-C64 were found to increase protein expression threefold [89]. Similarly, by introducing base substitutions into tRNA^Pyl^ conserved in human cytosolic tRNAs, tRNA^Pyl^ variants have been developed that exhibit enhanced UAG repression in eukaryotes [102].

To optimize the interactions between EF-Tu and Mj mutRNA^Tyr^_CUA_, a two-step rational evolution with rounds of negative and positive selection was performed to generate Tyr, p-azido-L-phenylalanine (pAzPhe), p-iodo-L-phenylalanine, p-acetyl-L-phenylalanine, p-benzoyl-L-phenylalanine, p-hydroxy-L-phenyllactic acid, L-bipyridylalanine, L-hydroxyquinolinylalanine, L-sulfotyrosine, p-azobenzyl-L-phenylalanine, o-nitrobenzyl-L-tyrosine and 7-hydroxycoumarinyl-L-ethylglycine [102]. In the first step, five base pairs of the T-strain were randomized (pairs 49–65, 50–64, 51–63, 52–62, and 53–61) and in the second step, eight bases in the acceptor arm that interacts with EF-Tu were randomized (2, 3, 6, 7, 66, 67, 70, and 71). As a result, the five best-designed tRNAs isolated bound EF-Tu 0.3–0.9 kcal/mol more tightly than the original tRNA, and the yield of GFP carrying the substitution at position Asn149 (AAU to UAG) was 3.8–11.8 times higher than that of the standard tRNA. The relative increase in protein yield was greatest for the ncAAs with the thickest side chains, reflecting their inherently lower affinity for EF-Tu. A similar strategy was used by Fan et al. [89], but with a triple randomization of Methanosarcina tRNA^Pyl^_CUA_ residues (first the 2–71 and 3–70 pairs, then the 6–67 and 7–66 pairs, and finally the 49–65 and 50–64 pairs) for incorporation into Nε-acetyl-L-lysine (AcLys) proteins. The most active variant carried the mutations C7G, C49U, C50G, G64C, G65A and G66C and contributed to a five-fold enhancement of the dual incorporation of AcLys into sfGPF in *E. coli*. This engineered tRNA^Pyl^ also increased the dual incorporations of Nε-Boc-L-lysine (BocLys), o-iodophenylalanine, TyrOMe, O-tert-butyl-L-tyrosine, o-cyano-L-phenylalanine, and o-nitro-L-phenylalanine by 1.4–2.7-fold.

Other approaches to tRNA engineering utilize the introduction of identity elements to obtain chimeric tRNAs and the introduction of mutations to improve tRNA stability and increase the efficiency of ncAA incorporation. One of the first examples of tRNA^Pyl^ engineering was the introduction of the A3G mutation in Mb tRNA^Pyl^, which allowed us to obtain an orthogonal system in S. cerevisiae cells—the native Mb-tRNA^Pyl^ carried the G3:U70 identity element for alanyl-tRNA synthetase [38]. Later, the U29aC substitution in the U29a:G41b pair was discovered to improve the rate of ncAA incorporation in HEK293T cells [103].

One of the challenges in developing archaeal tRNA^Pyl^ is that they have a secondary structure that is distinctly different from canonical and mammalian tRNAs, with similarity only to mitochondrial tRNA^Ser^_UGA_ [104]. Such a specific tRNA may be poorly compatible with the endogenous translation machinery and may be unstable in the cytosol. To overcome this difficulty, Serfling et al. [105] engineered novel variants of Mm-tRNA^Pyl^ and mt-tRNA^Ser^_UGA_ from *Bos taurus* that increase the yield of Lys (Boc) and Lys (Z) proteins in mammalian cells. In this case, mt-tRNA^Ser^_UGA_ was chimeric and carried elements of the tRNA^Pyl^ identity. The best variant Mm-tRNA^Pyl^ carried mutations in the D-stem—G11:C24 and G15 and G19; in the anticodon loop—C29; and in the T-loop—C56. An additional Ψ39 was also present, indicating the special role of modified bases in the efficiency of tRNA^Pyl^ and opening new horizons in its evolution. In mt-tRNA^Ser^_UGA_, the mutations of G73, G1:C72 and G2:C71 pairs in the acceptor stem; U8 at the junction of the acceptor stem and D-stem; G10:C25, C13:G22 and A14 pairs in the D-stem; G26 and the A31-U39 pair in the anticodon loop; C44, A45 and G48 in the variable loop; and G51:C63 in the T-stem were fixed. The resulting tRNAs had a 2–5-fold higher intracellular concentration compared to tRNA^Pyl^ carrying U29aC and were more stable due to the G19:C56 pair between the D- and T-loops.

When working with tRNAs that recognize UAGA-type quadruplets, an additional complication is competition with RF1. To overcome these limitations, modifications of the tRNAs are required to improve the translational efficiency of the quadruplets. DeBenedictis et al. [101] used PACE to isolate five tRNA variants (qtRNA^Gln^_UAGA_, qtRNA^Arg^_UAGA_, qtRNA^Ser^_UAGA_, qtRNA^Trp^_UAGA_ and qtRNA^Tyr^_UAGA_) that carried mutations in multiple regions of qtRNA, including regions that interact with cognate aaRS. In this case, the mutations did not result in the aminoacylation of the tRNA by unrelated synthetases, but qtRNA^Trp^_UAGA_ was predominantly misacylated with glutamine (81.7% Gln, 5.9% Trp, 12.4% Tyr), which was found to cause the A38U substitution, which is the identifying element of GlnRS [106]. Most mutations in qtRNAs were located in anticodon loops: for qtRNA^Arg^_UAGA_, for example, the mutation G45U was found in the anticodon stem; for qtRNA^Gln^_UAGA_—U31C in the anticodon stem; for qtRNA^Ser^_UAGA_—U32G, C33A, A40C, A41C in the anticodon stem and G53A in the variable; for qtRNA^Trp^_UAGA_—G24A in the D-loop, A38U in the anticodon stem and U72C in the acceptor leg; and for qtRNA^Tyr^_UAGA_—C34A and U35C in the anticodon stem.

More complex orthogonal tRNA/synthetase pairs have been constructed by designing chimeric tRNAs [107] and by the targeted mutagenesis of tRNAs for subsequent selection. Two typical examples of these strategies are tRNA UTu [108] and tRNA SecUX [109], constructed tRNAs that genetically encode selenocysteine. tRNA UTu was constructed as a hybrid between tRNA_Sec_ and tRNA_Ser_ that carries determinants for EF-Tu binding [108]. Due to the presence of tRNA_Sec_ segments, tRNA_Ser_ UTu is recognized by the SelA protein and converted into tRNA_Ser_ UTu. However, due to the tRNA_Ser_ segments, the tRNA Utu binds to EF-Tu and is delivered to the ribosome, which the natural tRNA_Sec_ is unable to do. Another tRNA for the genetic encoding of selenocysteine, SecUX tRNA, was derived from wild-type *E. coli* tRNA^Sec^_UCA_ and also selectively equipped with EF-Tu determinants [109]. These examples demonstrate that the successful design of tRNA/synthetase pairs usually requires the manipulation of both identity and anti-identity elements in the tRNA structure.

Furthermore, the recently identified class of tRNA molecules known as allo-tRNAs [110] and their successful use for selenoprotein synthesis [111] indicates that there may be many more natural tRNAs and tRNA-like molecules that can serve as templates for the future design of robust and useful orthogonal tRNA-synthetase/tRNA pairs for the expanded genetic code.

## 7. Mutually Orthogonal Systems

Over the last ten years, the methodology for incorporating ncAAs into proteins has developed considerably. However, the efficiency of incorporation can vary greatly and has been the subject of numerous optimization efforts. A particular challenge is the incorporation of multiple or different ncAAs into the same protein, which requires a sufficient number of orthogonal pairs. To solve this problem, the first mutually orthogonal pair was developed that allowed the incorporation of two separate ncAAs into the genetic code. Wang and colleagues used a wild-type or evolved PylRS-PylT UUA pair to repress the ochre stop codon and a Mj-TyrRS-Mj pair to repress the amber stop codon to sequentially incorporate two different non-canonical amino acids (ncAAs) into a single protein in *E. coli* with high efficiency [43]. Liu and colleagues have successfully demonstrated the simultaneous use of the Tyr (CUA) analog and the Pyl (UUA) analog [112]. Schultz and colleagues reported that the Pyl (UUA) analog and different variants of the Tyr (CUA) analog can simultaneously repress the amber and ochre codons in the GFP gene, with dual-tagged GFP yields ranging from 3 to 33% compared to the wild-type protein [113].

Another research group led by Meineke reported in 2023 on the dual incorporation of ncAAs into mammalian proteins. The scientists used mutually orthogonal PylRS/tRNA^Pyl^ pairs to suppress the opal and ochre stop codons in sfGFP and incorporate the amino acid pAzPhe at position 102 and N-ε-[(2-methyl-2-cyclopropen-1-yl)-methoxy]carbonyl-L-lysine (CpK) at position 150 during translation. Therefore, of all the cell lines selected for the study, the highest fluorescence signal was detected in the HCT116 line cultured in a CpK-containing medium [114].

In the search for mutually orthogonal systems, a new group of ΔPylSn was discovered, which led to the development of further PylRS/tRNA^Pyl^ variants capable of incorporating different ncAAs into a single translation system [59,66]. The Ma-PylRS enzyme from the ΔPylSn group showed a high UAG reading activity [60]. However, Ma-PylRS has been shown to be inferior in aminoacylation accuracy to enzymes of other groups, such as Mm-PylRS [33]. In this case, the high activity might not be related to the enzyme group but to its characteristics, such as a 5-fold higher solubility of Ma-PylRS due to the lack of a poorly soluble NTD [115]. It has also been shown that in organisms in which the ΔPylSn group is present, other PylRSs may also be present at the same time. For example, such a feature is found in the genome of *Methanomassiliicoccus luminyensis* (B10) and *Candidatus Methanohalarchaeum thermophilum* (HMET1) [116]. The two wild-type PylRS/tRNA^Pyl^ pairs from HMET1 have the following different identity elements: a motif loop 2 with a single amino acid deletion in PylRS2 and a discriminator base A73 in tRNA^Pyl^ 2. Based on these identity elements, Ma-tRNA^Pyl^ was designed with an extended variable loop that is orthogonal to the PylSn–PylSc group but still serves as a substrate for Ma-PylRS, an enzyme of the ΔPylSn group [59]. The combination of manipulated ΔPylSn and PylSn–PylSc enzymes enables the simultaneous incorporation of several different ncAAs into a single polypeptide [42,117].

One of the problems with inserting more than two ncAAs into a protein is the lack of an available codon. The intention was to use all three stop codons, but this strategy leaves no codon for termination, resulting in a toxic load on the host cell [118]. A new wave of research is based on GCE by introducing ncAAs in response to quadruplet codons. This strategy expands the possibilities of using both different and more than two ncAAs by increasing the variants of codons that do not encode natural amino acids [42].

Dunkelmann and colleagues created the first triple-orthogonal system based on the decoding of two quadruplet codons and an amber stop codon [19]. The scientists identified 12 sets of triple-orthogonal pairs: each triplet consists of one pair of Mm-PylRS tRNAs and two evolved pairs from the ΔPylSn group. Thus, each set of three triplet orthologous pairs contains enzymes such that each pair is orthogonal to each other and to the synthetases and tRNAs of *E. coli*. To efficiently incorporate ncAAs in response to quadruplet codons, the orthogonal ribosome ribo-Q1 was used to facilitate the decoding of quadruplet codons and amber codons [119].

Ohtsuki and colleagues succeeded in obtaining a triple-orthogonal quadruplet system. In their work, they presented the reprogramming of the CGGG, CUCU and GGGU codons for the incorporation of three different phenylalanine derivatives [120]. However, the use of quadruplet codons was previously only possible in single-cell systems. The development of an efficient four-letter decoding system instead of triplet codons in multicellular organisms will pave the way for a further expansion of the possibilities of ncAA incorporation [101]. In 2022, the triple incorporation of ncAAs into mammalian cells (HEK293T) was achieved using *E. coli* LeuRS/tRNA^Leu^_CUA_, *E. coli* TyrRS/tRNA^Tyr^_UUA_ and Mm-PylRS/tRNA^Pyl^_UCA_ [121].

Recent studies have shown an expansion in the use of PylRS and the corresponding tRNA. With three pairs of pyrrolysine tRNA synthetases and one pair of tyrosine tRNA synthetases from *Archaeoglobus fulgidus*, it is possible to incorporate four ncAAs into *E. coli* in vivo [117]. Recently, penta-orthogonal pairs have been proposed that can incorporate up to five ncAAs into the protein. A group of researchers led by Beattie showed experimental data on the cross-reactivity of PylRS/tRNA^Pyl^ to different groups to determine the thresholds for the sequence identity of mutually orthologous enzymes [40]. Thus, the study revealed five mutually orthogonal groups of PylRS: the canonical group N, corresponding to the fused group “PylSn–PylSc”; group S, corresponding to the separated group “PylSn+PylSc”; and ΔN PylRS lacking the N-terminal domain, comprising two subgroups A and B defined on the basis of differences in tRNA identity elements. The N and S groups are known to cross-react; interactions between the N, A and B groups can be eliminated by tRNA engineering. A set of PylRS/tRNA^Pyl^ pairs, with one pair from each of the five groups, was defined as the starting point for generating a set of quintuple-orthogonal pairs. This process led to the discovery of 924 reciprocally orthogonal pairs in 22 families, 1324 triply orthogonal pairs in 18 families, 128 quadruply orthogonal pairs in 7 families and 8 quintuply orthogonal pairs in 1 family. All newly generated PylRS/tRNA^Pyl^ sets matched or exceeded the orthogonality of all previous triple-orthogonal pairs. These advances, together with codon generation strategies that can be used to encode new amino acids, may facilitate the synthesis of proteins with increasing numbers of ncAAs.

## 8. Discussion

Recent studies have greatly expanded our understanding of the molecular basis for the orthogonality of PylRS/tRNA-Pyl pairs, opening new perspectives for the design and construction of tools to expand the genetic code. The aaRS/tRNA systems generated can efficiently express proteins with multiple native and unnatural amino acids, which is important for the further development of efficient orthogonal systems. In this area, significant progress has been made in the development of optimized aaRS/tRNA systems, accompanied by the discovery of different groups of synthetases capable of incorporating ncAAs into the protein structure. In addition, efforts are currently underway to generate more mutually orthogonal PylRS/tRNA-based pairs. The orthogonality of the constructed PylRS/tRNA^Pyl^ pairs has been confirmed in several living organisms, so the transplantation of key components from PylRS/tRNA^Pyl^ pairs into other paired systems is a promising approach for the development of new orthogonal translation systems. However, to improve the efficiency of genetic code expansion, we need to focus on increasing the solubility, aminoacylation activity and compatibility of tRNA^Pyl^ with ribosomes and other translational components in heterogeneous cells.

It is particularly important to understand how individual base substitutions in the tRNA structure affect specificity, as this may help in the further development of the targeted engineering of these molecules. Recent work has also highlighted the importance of the tRNA identity elements used by natural and engineered PylRS/tRNA^Pyl^ pairs to maintain mutual orthogonality, opening up the possibility of creating new pairs by combining different identity elements. This can lead to both increased specificity for the target amino acid and reduced cross-reactivity with other aaRSs, which in turn reduces the likelihood of errors in the translation process.

Active research in this area provides a basis for the modification of existing proteins and the creation of new ones. The genetic encoding of ncAAs may lead to new methods for controlling the secondary and tertiary structure of proteins as well as synthesizing proteins that are resistant to proteolysis, deepening the potential for scalable biosynthesis of modified therapeutic proteins. In the future, the combination of ncAA-coding strategies with the design of mutually orthogonal pairs and the generation of free codons will enable the biosynthesis of polymers composed entirely of ncAAs, with full control over their sequence and composition.

## Figures and Tables

**Figure 1 ijms-26-00539-f001:**
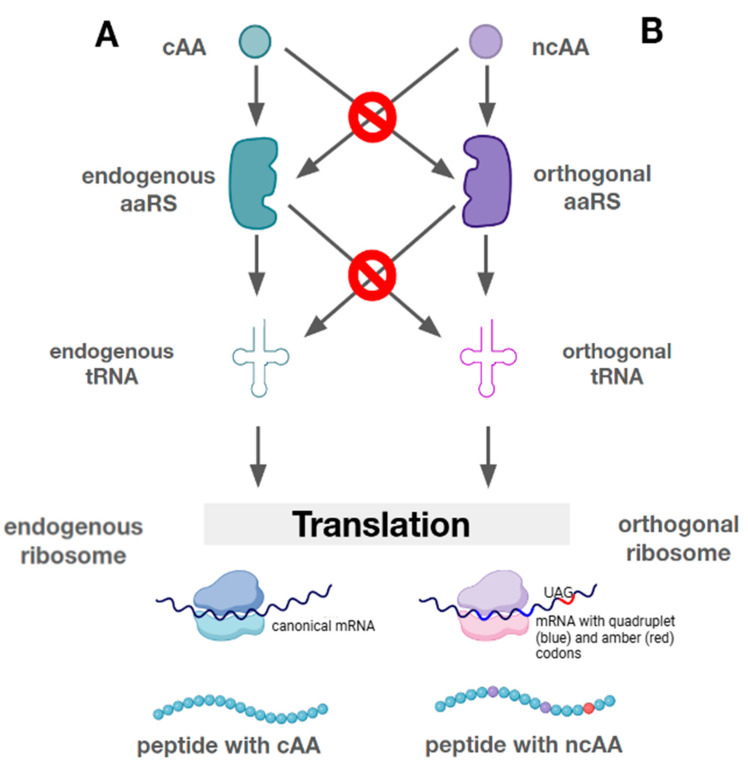
A comparison of natural and orthogonal translation systems. (**A**)—natural translation, which ends with the production of a peptide containing natural amino acids. (**B**)—orthogonal translation leading to the production of a peptide containing a non-proteinogenic amino acid that is incorporated when reading quadruplet codons (blue) or an amber stop codon (red). Crossed-out arrows indicate missing or minimal activity, which corresponds to the principle of orthogonality.

**Figure 2 ijms-26-00539-f002:**
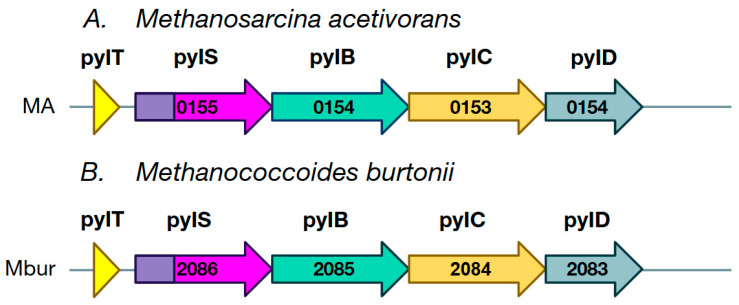
The structure of the Pyl cluster in the genomes of two representatives of the genera ((**A**)—*Methanosarcina*, (**B**)—*Methanococcoides*) of the *Methanosarcinaceae* family of methanogenic archaea. The number that marks each gene is the number of the locus in the annotated genome. PylRS (*pylS*) and tRNA^Pyl^ (*PylT*) are involved in the genetic coding of pyrrolysine. *pylB*, *pylC* and *pylD* are involved in the biosynthesis of pyrrolysine.

**Figure 3 ijms-26-00539-f003:**
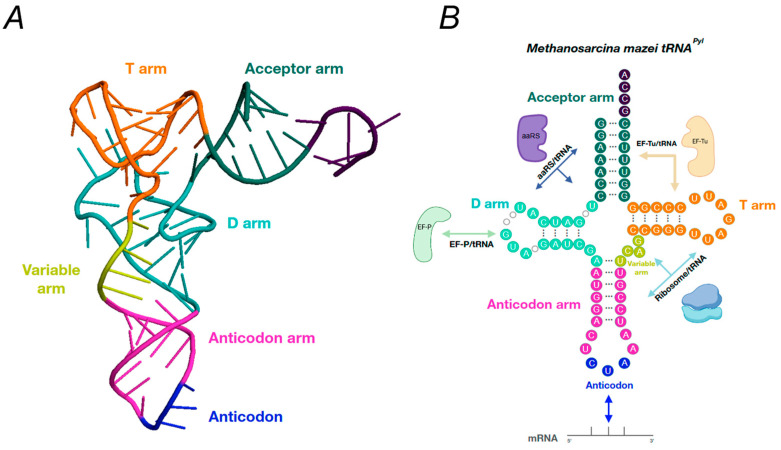
The structure of tRNA, with domains highlighted in different colors. (**A**) The three-dimensional structure of *Methanosarcina mazei* tRNA (PDB: 1N78) consists of the acceptor arm (green), the D-arm (light green), the anticodon arm (purple), the variable arm (yellow) and the T-arm (orange). (**B**) The secondary structure of the tRNA structure using the same color scheme, with the parts that interact with different factors. aaRSs and ribosomes interact with different parts of the tRNA, while EF-P interacts with the D-arm and elongation factor (EF-Tu) interacts with the T-stem and part of the acceptor stem.

**Figure 4 ijms-26-00539-f004:**
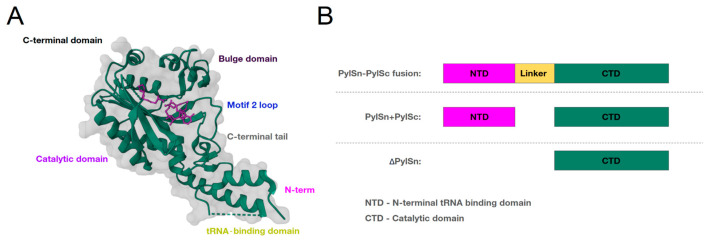
Domain organization of three groups of PylRS enzymes. (**A**): Structure of the N-terminal tRNA-binding domain of PylRS (PylSn) and the C-terminal catalytic domain (PylSc) from *Methanosarcina mazei*. (**B**): Domain organization of three groups of enzymes: “PylSn+PylSc”, “PylSn–PylSc” fusions, and “ΔPylSn”.

**Figure 5 ijms-26-00539-f005:**
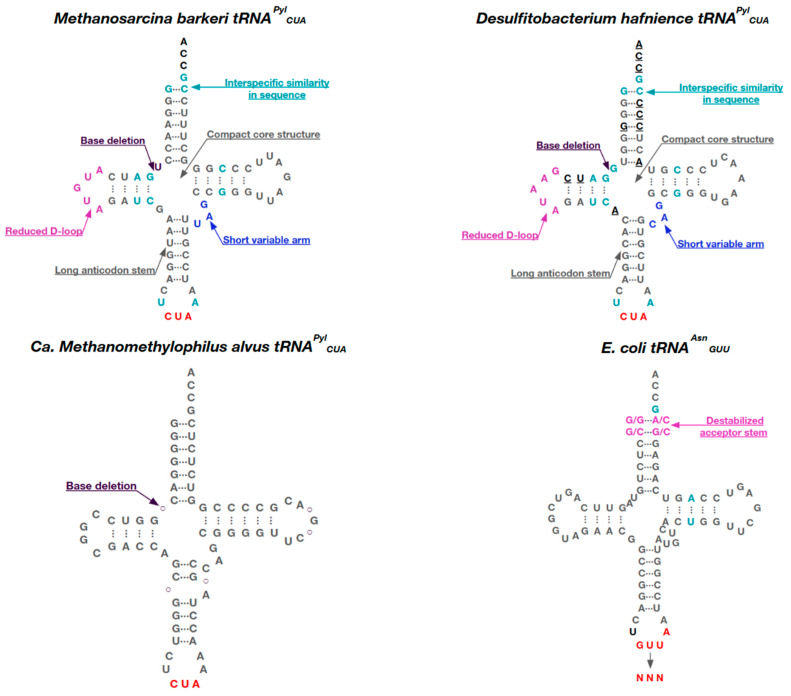
The structure of tRNA from different organisms—*Methanomethylophilus alvus*, *Methanosarcina barkeri*, *Desulfitobacterium hafnience* and *E. coli*. The figure shows the following characteristic structural features: nucleotides, stem and loop sizes, secondary structure fragment sizes, highly conserved regions that are the same in different species (marked in blue), unstable regions (marked in pink) and the anticodon region (marked in red).

## Data Availability

The original contributions presented in this study are included in the article/Appendix A. Further inquiries can be directed to the corresponding author.

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
