# Peer review of "Pyrrolysine Aminoacyl-tRNA Synthetase as a Tool for Expanding the Genetic Code"

_ijms, 2025, doi:10.3390/ijms26020539_

Round 1
Reviewer 1 Report
Comments and Suggestions for Authors
The review by Dakhnevich et al. summarizes knowledge about Pyrrolysyl-tRNA Synthetase and its use for gene coding expansion. The manuscript is not well structured and the need of this review as compared to recent reviews of the same topic by others is unclear. The authors should make a convincing effort to respond satisfactorily to major points 1 and 2.
MAJOR POINTS
1.- In the bibliography, I could not find any reference to work by the authors of the manuscript. I wonder why, in the absence of previous experience in the field, they decided to review this topic. In other words, the readers will wonder what are the credentials of the authors that support their knowledgeability. In my opinion, having research experience in the reviewed topic is the major credential needed. This question should be carefully justified in the introduction of the review.
2.- Searching PubMed for “Pyrrolysyl-tRNA Synthetase[Title] OR pyrrolysine[Title]” recovered 13 review articles written in English published in 2002–2024. One of these was published in 2023 and two in 2024. They are cited as references 34, 41 and 51, not as reviews of the topic, but as support of very specific points. The introduction of the manuscript should mention explicitly the occurrence of at least these recent reviews, and should specifiy what makes the current review manuscript by Dakhnevich et al. useful and unique, with respect to references 34, 41 and 51.
3.- There are several aspects of nomenclature that have to be addressed:
a) There are important abbreviations that need definition or their definition is misplaced in the manuscript, for instance: PylRS, Mm-PylRS, Mb-PylRS, Ma-PylRS, PylSn-PylSc, ncAA, cAA, aaRS, ARS, etc etc. The whole manuscript should be revised concerning abbreviations used and their definitions. An abbreviation list would be useful, rather than defining in the text.
b) The concept of enzyme classes is ill defined. See for instance its use in lines 13, 85, 94, 150–165, 208–290, etc etc. There should be a clear differentiation between class I/II aminoacyl-tRNA synthetases (mainly used in section 2) and the different "classes" of PylRS enzymes (mainly used in posterior sections). In the latter case another term could be used (perhaps "type", "kind" or "variant").
c) Tritrpticin is misspelled in lines 51–54.
d) The concept of "free codon" should be explained (line 71).
MINOR POINTS
The structure and biological distribution of pirrolysin and PylSR enzymes should be described in the introduction.
Taxonomical names should be in italics (e.g. lines 123, 125, 197)
Line 160, insert "universal" after "last".
Author Response
MAJOR POINTS
Comment 1: In the bibliography, I could not find any reference to work by the authors of the manuscript. I wonder why, in the absence of previous experience in the field, they decided to review this topic. In other words, the readers will wonder what are the credentials of the authors that support their knowledgeability. In my opinion, having research experience in the reviewed topic is the major credential needed. This question should be carefully justified in the introduction of the review.
Response 1: Our team decided to write this review publication after we had conducted our own research in the field of genetic code reprogramming. We are working on a project related to site-directed mutagenesis of pyrrolysine aminoacyl-tRNA synthetases (PylRS) using the phage-assisted continuous evolution (PACE) method. We have engineered several novel PylRS variants capable of integrating unnatural amino acids using the amber stop codon. Our team has already obtained convincing results showing that our mutant enzymes can incorporate a range of unnatural amino acids into proteins via biosynthesis. Experimental data are currently being finalized. We expect that our experimental article will be published in the first quarter of 2025. Since a large amount of literature data was processed, we have decided to publish the review article before the experimental part is finalized. We hope that this review will be useful for the community and provide valuable information for further research in this area.
Comment 2: Searching PubMed for “Pyrrolysyl-tRNA Synthetase[Title] OR pyrrolysine[Title]” recovered 13 review articles written in English published in 2002–2024. One of these was published in 2023 and two in 2024. They are cited as references 34, 41 and 51, not as reviews of the topic, but as support of very specific points. The introduction of the manuscript should mention explicitly the occurrence of at least these recent reviews, and should specifiy what makes the current review manuscript by Dakhnevich et al. useful and unique, with respect to references 34, 41 and 51.
Response 2: In a recent review by Daniel L. Dunkelmann (citation 41), authors discuss the design and optimization of PylRS/tRNAPyl pairs in terms of enzyme evolution to expand the genetic code in E. coli and other prokaryotic and eukaryotic systems. In contrast to our review, Dunkelmann et al. do not analyze the effects of specific mutations on the ability of PylRS to incorporate ncAAs, but rather provide a comprehensive summary of evolutionary methods for enzyme engineering. In addition, a large part of the review is devoted to applications of ncAAs that can currently be incorporated into proteins via biosynthesis. Dunkelmann et al. have convincingly shown that not only unnatural amino acids, but also small molecules that do not belong to amino acids, can be used for this purpose. It is important to emphasize that Dunkelmann et al. were the first to introduce new nomenclature for the groups of aminoacyl-tRNA synthetases and created triply- and even penta-orthogonal systems based on their mutual orthogonality, which is very respectfully cited in our paper.
The review paper by Nikolaj G Koch et al. (now citation 53), as the authors noted themselves, is primarily devoted to all the substrates that can be incorporated into proteins using PylRS and the applications for which these ncAAs can be used, with a focus on therapeutic applications. Koch et al. summarize results of previous studies in the field of enzyme engineering methods and propose, in their opinion, the best system for the evolution of PylRS with improved in vivo efficiency for the recognition of new substrates. In contrast, our review addresses this issue indirectly and focuses on specific mutations in different domains of PylRS that can improve the aminoacylation of ncAAs. Although the focus of our review is on PylRS, we also briefly address another major issue in orthogonal translation: the limited number of ncAAs that can be incorporated simultaneously and the need to increase the number of codons (apart from stop codons) used for the incorporation of ncAAs and the limited number of ncAAs that can be incorporated simultaneously, which are also addressed in our work.
A review paper by Xuemei Gong et al. (citation 34) provides a great overview of the key features of the PylRS/tRNAPyl pair that make this tool particularly suitable for genetic code expansion. In particular, their work focuses on important developments to improve the efficiency and (mutual) orthogonality of PylRS/tRNAPyl for cotranslational and site-specific incorporation of multiple ncAAs. It should be noted that our review addresses a similar topic: the structural and functional properties of native and engineered PylRS/tRNAPyl pairs. However, Gong et al. address this topic from the perspective of new ncAA substrates and recent advances in the incorporation of different ncAAs using orthogonal pairs as well as their optimization in terms of efficiency and substrate specificity.
Our team has a slightly different goal: to update information related to effects of different mutations on the ability to incorporate unnatural amino acids into proteins using PylRS. We have carefully selected information on important mutations in the amino acid sequences of PylRS that improve the catalytic properties of the enzyme and increase its selectivity for certain groups of unnatural amino acids. One of the most important aspects of our review is the update of the data on substitutions in the N-domain of the enzyme that not only improve its properties but can also be applied to PylRS of other groups with an N-domain. Another important advantage of our review compared to others is the chapter dedicated to tRNA engineering in the context of improving binding to ncAAs and mutant PylRS, which distinguishes our work from other reviews.
Comment 3: There are several aspects of nomenclature that have to be addressed:
Comment 3.a: There are important abbreviations that need definition or their definition is misplaced in the manuscript, for instance: PylRS, Mm-PylRS, Mb-PylRS, Ma-PylRS, PylSn-PylSc, ncAA, cAA, aaRS, ARS, etc etc. The whole manuscript should be revised concerning abbreviations used and their definitions. An abbreviation list would be useful, rather than defining in the text.
Response 3.a: Thank you for pointing this out. We have added a list of abbreviations at the end of work to aid understanding and updated abbreviations in the text.
Comment 3.b: The concept of enzyme classes is ill defined. See for instance its use in lines 13, 85, 94, 150–165, 208–290, etc etc. There should be a clear differentiation between class I/II aminoacyl-tRNA synthetases (mainly used in section 2) and the different "classes" of PylRS enzymes (mainly used in posterior sections). In the latter case another term could be used (perhaps "type", "kind" or "variant").
Response 3.b: We have changed the word classes to groups where we previously referred to “classes” of the PylRS enzyme.
Comment 3.c: Tritrpticin is misspelled in lines 51–54.
Response 3.c: Thank you, we have corrected this mistake.
Comment 3.d: The concept of "free codon" should be explained (line 71).
Response 3.d: We have added an explanation of the concept of a free codon in lines 65-66.
MINOR POINTS
Comment 4.a: The structure and biological distribution of pyrrolysine and PylSR enzymes should be described in the introduction.
Response 4.a: We discuss the structure and groups of the enzyme in the corresponding chapters so as not to overload the text of the introduction, which is devoted to the expansion of the genetic code and the role of PylRS. However, we agree with your suggestion and have added several statements at the end of the Introduction section.
Comment 4.b: Taxonomical names should be in italics (e.g. lines 123, 125, 197)
Response 4.b: Thank you, we have changed all taxonomic names to italics.
Comment 4.c: Line 160, insert "universal" after "last".
Response 4.c: Thank you for this correction, we have inserted this word.

Reviewer 2 Report
Comments and Suggestions for Authors
The manuscript of Dakhnevich et al is a review that reports recent studies on pyrrolysine synthetases (PylRS). In particular, the authors focused on correlation between structure and activity of PylRS and tRNAPyl and the use of pyrrolysine synthetases to extend the genetic code.
In my opinion the manuscript is of interest but some points reported below need to be clarified.
1- First of all, the manuscript pdf has some pagination problems on the first page. Please correct this.
2- If there are kinetic data on these synthetases please include them in the manuscript.
3- The legend of figure 1 shows the division of the image into two sectors, called A and B. The letters A and B are not reported in the image.
4- page 4, line 121. I advise the authors to also mention the twenty-first amino acid.
5- The reported files indicated on page 10 are missing in the supplementary materials.
6- Please confirm writing amino acid residues with or without hyphen between amino acid name and sequence number.
Author Response
Comment 1: First of all, the manuscript pdf has some pagination problems on the first page. Please correct this.
Response 1: We have attached an updated version of the PDF file without the pages that have shifted. We apologize for the inconvenience caused by the pagination problem.
Comment 2: If there are kinetic data on these synthetases please include them in the manuscript.
Response 2: Thank you for this valuable suggestion. We have provided kinetic data for some enzyme variants with native substrate and non-natural amino acids. These data are listed in Section 4.3 and in Supplementary Table 1.
Comment 3: The legend of figure 1 shows the division of the image into two sectors, called A and B. The letters A and B are not reported in the image.
Response 3: Thank you very much, we have corrected the figure.
Comment 4: page 4, line 121. I advise the authors to also mention the twenty-first amino acid.
Response 4: Thank you very much for this suggestion. We have added a short summary about the 21st unnatural amino acid selenocysteine on page 4 lines 131-134.
Comment 5: The reported files indicated on page 10 are missing in the supplementary materials.
Response 5: We are sorry, but we have not found the reference to supplementary materials on page 10.
Comment 6: Please confirm writing amino acid residues with or without hyphen between amino acid name and sequence number.
Response 6: Thank you very much for this correction. We rechecked the entire text for errors and corrected the spelling of amino acid substitutions without a hyphen between the amino acid and its position in the protein.

Round 2
Reviewer 1 Report
Comments and Suggestions for Authors
The revised version of the manuscript by Dakhnevich et al. gives only partial response to major points 1 and 2, to which they were asked to “make a convincing effort to respond satisfactorily”.
MAJOR POINT 1
In the bibliography, I could not find any reference to work by the authors of the manuscript. I wonder why, in the absence of previous experience in the field, they decided to review this topic. In other words, the readers will wonder what are the credentials of the authors that support their knowledgeability. In my opinion, having research experience in the reviewed topic is the major credential needed. This question should be carefully justified in the introduction of the review.
The authors give a detailed response to this point in their cover letter, but this is not considered in the revised version of the manuscript. Therefore, upon reading the manuscript, one still wonders what are the credentials of the authors that support their knowledgeability. Can they devote a short introductory paragraph to their work in progress? If not, they could postpone the publication of this review until they can cite their own results.
MAJOR POINT 2
Searching PubMed for “Pyrrolysyl-tRNA Synthetase[Title] OR pyrrolysine[Title]” recovered 13 review articles written in English published in 2002–2024. One of these was published in 2023 and two in 2024. They are cited as references 34, 41 and 51 (now 53), not as reviews of the topic, but as support of very specific points. The introduction of the manuscript should mention explicitly the occurrence of at least these recent reviews, and should specifiy what makes the current review manuscript by Dakhnevich et al. useful and unique, with respect to references 34, 41 and 51 (now 53).
The cover letter gives much detail about this request but it has not been considered in the revised manuscript.
NEW MINOR POINTS
In the keywords, “pares” should be changed to “pairs”.
In Figure 1, the type of the lettering adjacent to the ribosomes is too small.
The titles of section 4.1 and 4.2 need amendment.
The Supplementary Table 1 needs careful formatting. A horizontal rather than vertical design may be helpful. Full references should be cited, not just DOIs.
Author Response
MAJOR POINT 1
In the bibliography, I could not find any reference to work by the authors of the manuscript. I wonder why, in the absence of previous experience in the field, they decided to review this topic. In other words, the readers will wonder what are the credentials of the authors that support their knowledgeability. In my opinion, having research experience in the reviewed topic is the major credential needed. This question should be carefully justified in the introduction of the review.
The authors give a detailed response to this point in their cover letter, but this is not considered in the revised version of the manuscript. Therefore, upon reading the manuscript, one still wonders what are the credentials of the authors that support their knowledgeability. Can they devote a short introductory paragraph to their work in progress? If not, they could postpone the publication of this review until they can cite their own results.
MAJOR POINT 2
Searching PubMed for “Pyrrolysyl-tRNA Synthetase[Title] OR pyrrolysine[Title]” recovered 13 review articles written in English published in 2002–2024. One of these was published in 2023 and two in 2024. They are cited as references 34, 41 and 51 (now 53), not as reviews of the topic, but as support of very specific points. The introduction of the manuscript should mention explicitly the occurrence of at least these recent reviews, and should specifiy what makes the current review manuscript by Dakhnevich et al. useful and unique, with respect to references 34, 41 and 51 (now 53).
The cover letter gives much detail about this request but it has not been considered in the revised manuscript.
Response 1: We understand the importance of clarity and validity in scientific publications, so we have made changes to the text of the introduction section. Features of this manuscript that distinguish it from other reviews are described in detail in the new version. We have also mentioned our ongoing experimental research in the introduction. Although its results have not yet been published, we believe that this review will be useful for the scientific community. Thank you for your rigorous review and for the corrections you made. They helped to significantly improve the quality of the manuscript.
NEW MINOR POINTS
In the keywords, “pares” should be changed to “pairs”.
Response 2: Thank you very much, we have corrected this mistake.
In Figure 1, the type of the lettering adjacent to the ribosomes is too small.
Response 3: Thank you, we have corrected the figure.
The titles of section 4.1 and 4.2 need amendment.
Response 4: Thank you for pointing this out. We have changed the titles.
The Supplementary Table 1 needs careful formatting. A horizontal rather than vertical design may be helpful. Full references should be cited, not just DOIs.
Response 5: Thank you very much for this suggestion. We have changed the design of the table in supplementary materials.
Reviewer 2 Report
Comments and Suggestions for Authors
The authors have clarified my doubts and answered my questions. In my opinion the manuscript is acceptable for publication if the editor agrees.
Author Response
Comment 1: The authors have clarified my doubts and answered my questions. In my opinion the manuscript is acceptable for publication if the editor agrees.
Response 1: Thank you for your thorough review and corrections, which greatly enhanced the quality of the manuscript.
Round 3
Reviewer 1 Report
Comments and Suggestions for Authors
I thank the authors because the major points raised by this reviewer are satisfactorily solved in the second revised version.
On the other hand, two of the minor points are still pending despite authors’ claims. Namely:
The small type of Figure 1 has NOT been changed.
The titles of sections 4.1 and 4.2 have NOT been corrected. Please, note as these titles read “C. hanges” rather than “Changes”.
Author Response
Comment 1:
On the other hand, two of the minor points are still pending despite the authors' claims. Namely:
The small type of Figure 1 has NOT been changed.
The titles of sections 4.1 and 4.2 have NOT been corrected. Please, note as these titles read “C. hanges” rather than “Changes”.
Response 1: Thank you for pointing this out. We apologize for any inconvenience caused. All minor points have been addressed in the updated version of the manuscript.